# Annihilation of action potentials induces electrical coupling between neurons

**Moritz Schloetter[1,2]\*, Georg U Maret[1], Christoph J Kleineidam[2]**

[1]Department of Physics, University of Konstanz, Konstanz, Germany; [2]Neurobiology, Department of Biology, University of Konstanz, Konstanz, Germany

## eLife Assessment

This **important** study enhances our understanding of ephaptic interactions by utilizing earthworm recordings to refine a general model and use it to predict ephaptic influences across various synaptic configurations. The integration of experimental evidence, a robust mathematical framework and computer simulations **convincingly** demonstrate the effects of action potential propagation and collision properties on nearby membranes. The study will interest both computational neuroscientists and physiologists.

**\*For correspondence:**
moritz.schloetter@uni-konstanz.de

**Competing interest:** The authors declare that no competing interests exist.

**Abstract** Neurons generate and propagate electrical pulses called action potentials which annihilate on arrival at the axon terminal. We measure the extracellular electric field generated by propagating and annihilating action potentials and find that on annihilation, action potentials expel a local discharge. The discharge at the axon terminal generates an inhomogeneous electric field that immediately influences target neurons and thus provokes ephaptic coupling. Our measurements are quantitatively verified by a powerful analytical model which reveals excitation and inhibition in target neurons, depending on position and morphology of the source-target arrangement. Our model is in full agreement with experimental findings on ephaptic coupling at the well-studied Basket cell-Purkinje cell synapse. It is able to predict ephaptic coupling for any other synaptic geometry as illustrated by a few examples.

## Introduction

Neurons are characterized by electrically excitable membranes and most neurons in a brain are able to generate action potentials (APs) which are propagating along the axon. At the synaptic terminal they eventually lead to electrochemical processes, allowing communication with postsynaptic target neurons (*Eccles, 1982*). The most prevalent form of communication between neurons is by transmitter release at the presynaptic terminal, and subsequent binding of these at postsynaptic sites leads to modulation, e.g., excitation or inhibition. Less common in nervous systems of any organism are gap junctions between neurons, resulting in an aggregated functional unit because internal current is transferred from one neuron to the other neuron.

Beside these two forms of communication between neurons, the electrical field itself which is generated at an excited membrane can have an impact on the membrane potential of cells in the vicinity (*Jefferys, 1995*; *Anastassiou and Koch, 2015*; *Rebollo et al., 2021*). Rapid changes of the membrane potential, as it occurs when APs are generated, induce electrical fields, that by superposition are also present in neighboring cells. The impact of a single, propagating AP on adjacent neurons is very small, nevertheless, along active, parallel fibers synchronization of APs can occur (*Katz and Schmitt, 1940*). When the term ephaptic interaction was coined by *Arvanitaki, 1942*, she concluded that the blockage of APs is the main source of such effects (hereafter ephaptic coupling) in nervous

systems. Indeed, APs arriving at the synaptic terminal of a neuron annihilate, thereby generating a more potent electrical field for ephaptic coupling. The functional significance of ephaptic coupling at synaptic terminals is well documented in at least two systems: at the Mauthner cell in teleost fish and at the Purkinje fibers of the cerebellum in vertebrates (*Furukawa and Furshpan, 1963*; *Blot and Barbour, 2014*). In both systems, APs arriving at the synaptic terminal of the source neuron (e.g. the Basket cell of the cerebellum) modulate the initiation of an AP in the target neuron by transiently changing the membrane potential. Although the phenomenon of ephaptic coupling is known since many decades (*Arvanitaki, 1942*), a quantitative physical description is missing. This is even more surprising when considering that ephaptic coupling at synaptic terminals is ubiquitous and may well contribute to communication between many different types of neurons, beside the mentioned Mauthner cell and Purkinje fiber.

In our study, we aim to describe the electric field of an AP arriving at the synaptic terminal in such detail that the ephaptic coupling and the change in membrane potential of a known target neuron can be predicted.

The propagating AP is driven by a voltage-dependent transition of the membrane from a resting to an excited state. As a consequence of electroneutrality, current flows across the membrane resulting in a closed loop through extracellular and intracellular medium. The current loop includes the resistive extracellular medium where the AP generates a voltage signature. In 1952, Hodgkin-Huxley introduced a model (HH model) for generation and propagation of APs (*Hodgkin and Huxley, 1952*). It is based on the cable equation and incorporates Nernst equations for different ions and changing ion-specific conductances upon activation of ion channels.

The HH model is a powerful tool to study and evaluate the properties of ion channels in excitable membranes on a microscopic level, and the great attention it received since being introduced is more than justified (*Catterall et al., 2012*). However, the macroscopic phenomenon of current loops and associated electric fields is already well accounted for by a much simpler model introduced by Tasaki and Matsumoto (TM model) (*Tasaki and Matsumoto, 2002*; *Tasaki, 2006*) without the need of a multitude of specific parameters as required for example in the HH model. While the TM model can be considered as a simplification of the HH model, the mechanism that the authors had in mind is very different (*Tasaki, 2002*; *Tasaki et al., 1965*; *Tasaki et al., 1971*).

In general terms, neural membranes can go through transitions where the electric properties change extremely fast (*Fedosejevs and Schneider, 2022*; *Mussel et al., 2021*; *Horkay et al., 2000*). The TM model describes a resting and an excited state, each with a linear cable model. It uses the cable equation with two state-specific parameters, its equilibrium potential and conductivity, which undergo a sudden (voltage-dependent) switch at the boundary between resting and excited state. The boundary between an excited and a resting section propagates toward the resting side. The AP in the TM model has only 3 degrees of freedom, e.g., its propagation velocity, its amplitude, and its length. Nevertheless, the TM model is well suited to describe the current distribution that is driven by the inhomogeneity of local electric fields in the internal and the external medium. Thus, the TM model is already sufficient to describe the mutual influence of neurons by ephaptic coupling, using only three parameters for each neuron.

It is therefore of prime importance to evaluate the quantitative description of the TM model based on experimentally derived values. We explore the behavior of APs at boundaries to assess these three parameters. A suitable boundary condition (intracellular, axial current equals zero) can be generated experimentally by a collision of two counter-propagating APs (*Tasaki, 1949*; *Spach et al., 1971*; *Shrivastava, 2018*; *Shrivastava et al., 2018*; *Shrivastava et al., 2018*). Within any cable model, the very same boundary condition also exists within the axon at the synaptic terminal due to the broken translation symmetry for the current loops (*Spach and Kootsey, 1985*).

In this study, we first focus on collisions of APs. Our experimental observation of colliding APs provides unique access to the spatial profile of the extracellular potential around APs that are blocked by collisions and thus annihilate. Our procedure is as follows: Recording propagating APs allows to determine both the propagation velocity and the amplitude of the extracellular electric potentials. The collision experiment provides additional information on the characteristic length $\lambda^\star$, thereby fully determining the parameters of the TM model. We find that TM is sufficient to describe the onset of the electric field dynamics of an AP. However, as TM does not include the slow return of the excited membrane to the resting state, we introduce an extension of the TM model that we call the relaxing

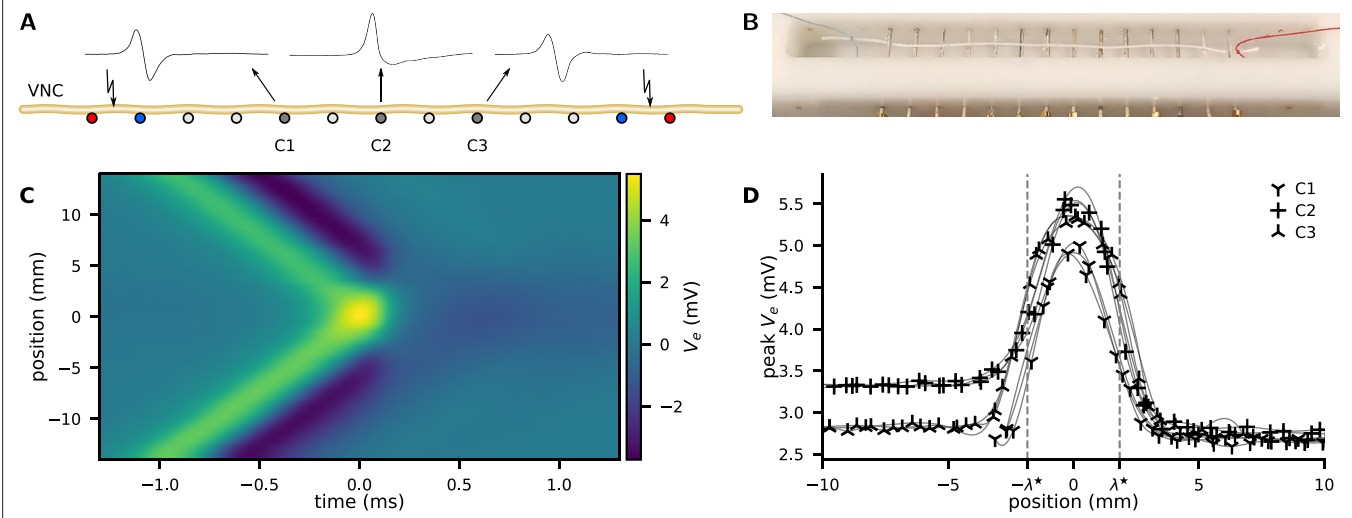

**Figure 1.** Action potential (AP) collision experiment. (**A**) The ventral nerve cord (VNC) of an earthworm is positioned on a series of transverse, single-ended electrodes to excite and monitor APs. Individual or two counter-propagating APs are generated by stimulating the nerve at the outermost electrodes. Some of the inner electrodes (C1, C2, C3) are used for recording; all other electrodes are grounded. Propagating APs generate biphasic electrical potential as sketched above C1 and C3 whereas colliding APs provoke an essentially unipolar peak (cf. C2). By variation of the delay between the opposing stimuli, the collision can be generated anywhere along the nerve between the stimulation electrodes. (**B**) Photograph of the recording chamber with a white thread to illustrate the position of the VNC. Distance between the electrodes is 5 mm. (**C**) Extracellular recording of propagating and colliding APs. A collision sweep experiment yields multiple recordings with varying distance between recording site and point of collision. The collision is captured in the recording line at $y$-position 0 mm, while orthodromic propagation is at the top and antidromic propagation is at the bottom. (**D**) The peak amplitude as a function of the distance to the collision. Examples of four sweeps at three positions along the nerve cord. As a guide to the eye, the data points are connected by a cubic spline (thin lines).

The online version of this article includes the following figure supplement(s) for figure 1:

**Figure supplement 1.** Complete raw data from experiment number 3 (out of three).

**Figure supplement 2.** Gray crosses show the maximal negative deflection of the extracellular potential from 88 recordings (C1, C2, C3 from three experiments).

Tasaki model (RTM) which adds an empirical slow relaxation term to the TM model. We find that RTM accounts for the full spatiotemporal signature of propagating and colliding APs and is therefore also very useful to predict electric effects at axon terminals, e.g., at synapses. Since the RTM sets a complete framework to describe APs and to predict ephaptic interactions for a given morphology, the RTM is used in the last part of this paper to simulate various representative pre- and postsynaptic morphologies and geometries. Around AP propagation boundaries, we find excitatory as well as inhibitory regions, depending on timing, relative position, orientation, and morphology of source and target neuron. We expect these predictions to be robust, because the model is absolutely minimal and matches the experimental observations.

## Results

### Extracellular potential around AP collisions

We record the extracellular electric potential of APs, generated by the median giant fibers (MGF) of the ventral nerve cord (VNC) of an earthworm using the setup shown in *Figure 1A and B*. In a first experiment the ends are stimulated individually and the APs are detected at several positions along the fiber. We verify the vitality of a nerve cord by asserting that both fibers (MGF and the lateral giant fibers [LGF]) reliably conduct APs in both directions, and in our experiments, we investigate selectively the MGF only.

The signatures $V_e(t)$ of both orthodromic and antidromic propagating APs look indeed very similar. We observe a biphasic voltage spike propagating with 14.5(11) m/s (*Figure 1A*).

In the main experiment, both ends are stimulated simultaneously and the APs collide close to the central recording electrode. We observe that the APs do not penetrate each other but always

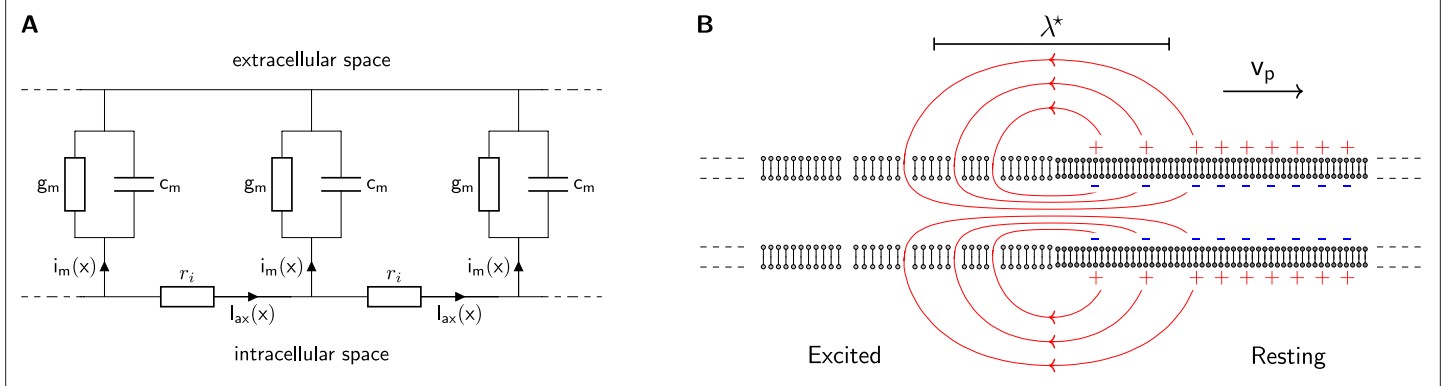

**Figure 2.** The classical cable model. (**A**) A chain of identical RC circuits with conductivity $g_m$, capacity $c_m$, and inner resistivity $r_i$ mimics the electric properties of the cellular membrane and connects the extracellular and intracellular space. The change of inner axial current $I_{ax}(x)$ is given by the transmembrane current density $i_m(x)$ (Kirchhoff's current law). (**B**) The propagating solution of the Tasaki-Matsumoto model. A boundary between a resting state and an excited state induces an axial current, that causes a propagation of the boundary. The axial current flows in closed loops and returns within the resistive extracellular medium, causing an extracellular potential that travels with the current source along the neuron.

annihilate at the collision site (*Figure 1A*; *Tasaki, 1949*). At this position, the trace of $V_e(t)$ becomes more monophasic, with an almost doubled positive peak amplitude (*Figure 1A*). The negative phase of $V_e(t)$ at the collision site is considerably diminished and shows a distinct slow relaxation. The peak amplitude of propagating APs is 2.9(2) mV while the colliding APs peak is 5.2(3) mV.

The width of the collision is a measure of the characteristic length $\lambda^\star$ of the AP and is uniquely revealed by a collision sweep experiment. We experimentally positioned the site of collision along the neuron by introducing a delay between the opposing stimuli. Assuming symmetric propagation velocities, a delay $\Delta t$ displaces the collision by $x = v_p \Delta t / 2$, where $v_p$ is the propagation velocity and $x$ is the distance between the collision and the recording electrode. The spatial extend of the collision process is then found by repeating the experiment at various delays $\Delta t$ between the opposing stimuli, see *Figure 1C*. The propagation velocity is derived from the time of arrival at separate recording sites. The peak amplification can be used as a measure of the width of the collision (*Figure 1D*). We use four recordings at three positions (anterior, medial, and posterior) along the nerve cord and we assess the mean full width at half maximum as 3.8(5) mm.

## Model of the AP

We describe a neuron by a classical cable model as shown in *Figure 2A* (see 'Materials and methods' for further details). Such a model consists of a chain of RC circuits composed of resistors with conductivity $g_m$(S/m) and capacitors $c_m$(F/m) which are connected by an inner resistivity $r_i(\Omega/m)$. The inner axial current $I_{ax}$ is driven by the gradient of the potential $V(x, t)$ inside the neuron. A common approach is to assume a constant capacity and to neglect external fields and inhomogeneities in the neuron. Then, the balance of currents results in the cable equation with the membrane potential $V_m$:

$$\frac{\partial^2 V_m}{\partial x^2} = r_i c_m \frac{\mathrm{d} V_m}{\mathrm{d} t} + r_i g_m V_m \tag{1}$$

The first term on the right describes the capacitive charging by an axial current which is linked to the rate of change in membrane potential. The second term describes the contribution of resistive membrane current to axial current. The relationship between the axial current $I_{ax}$ and the membrane current $i_m$ is further described in the Materials (*Equation 5*).

APs are characterized by a solitary, i.e., non-spreading spatiotemporal shape. In the cable model, the generation of such APs requires an additional nonlinear response of the membrane. Commonly, living cells are electrically charged to a negative equilibrium potential $V_{eq}$ across their membrane. In excitable cells, e.g., neurons, a change of the membrane potential beyond a critical threshold value leads to a rapid transition from the resting into an excited state of the membrane which essentially differ in their respective resistivities. The excited state, with drastically increased conductivity, is indicated by the starred values $g_m^\star$ and potential $V_{eq}^\star$.

A local transition of a membrane from the resting to excited state results in a boundary between $V_{eq}$ and $V_{eq}^{\star}$. The difference in membrane potentials between the resting and excited state induces an internal axial current. The increased conductivity at the excited state *drives* the internal current and depolarizes the adjacent membrane. The flow lines of internal, transmembrane, and external current form a closed loop around the position of the transition as sketched in *Figure 2B*. All internal current is present as an extracellular return current and in neurons, this process produces a propagating, dipolar local current source.

The TM model describes both the resting and the active state by linear cable models (*Tasaki and Matsumoto, 2002*; *Tasaki, 2006*). The resting state has a negative electrochemical equilibrium potential $V_{eq} = -100\,\mathrm{mV}$ and a very low conductivity $g_m$. In the excited state the conductivity is drastically increased to $g_m^{\star} \approx 100\,g_m$ and the equilibrium potential $V_{eq}^{\star}$ is zero. The transition from the resting to the excited state is discontinuous and occurs at $V_m = -50\,\mathrm{mV}$. The TM model does not address how the resting state is maintained before a transition, nor does it address how the resting state is restored thereafter.

A simple analytical solution of the TM model describes the propagating initial depolarization of a nerve signal. This includes an exact expression for the propagation velocity $v_p$ and for the spatial length $\lambda^{\star}$ of the initial current dipole as illustrated in *Figure 2B*. The velocity is determined by the delay in charging of neighboring capacitors, while the leak conductivity barely effects the process. Therefore, we base our study on simplified expressions by neglecting the leak conductivity, as it amounts to less than 1% of the conductivity in the excited state (*Tasaki and Matsumoto, 2002*). Then, the analytical solution of the TM model yields

$$v_p = \frac{1}{c_m}\sqrt{\frac{g_m^{\star}}{2\,r_i}} \qquad \text{and} \qquad \lambda^{\star} = \frac{1}{\sqrt{r_i\,g_m^{\star}}} \tag{2}$$

Note that the length scale $\lambda^{*}$ used here is not equivalent to $\lambda$ used in basically all previous models where $\lambda$ denotes the damping length due ionic leakage neglected here. The products in *Equation 1* can be expressed as $r_i c_m = \left(v_p \lambda^{\star}\right)^{-1}$ and $r_i g_m^{\star} = \lambda^{\star -2}$. According to *Equation 1* $(r_i c_m)^{-1}$ is the diffusion constant of the spreading potential $V_m$ which connects a typical timescale $\tau_L$ with a typical diffusion length $L$ by $\tau_L = r_i c_m L^2$. The two products $r_i c_m$ and $r_i g_m^{\star}$ are directly determined in our experiment from measurements of $v_p$ and $\lambda^{\star}$, and for a comparison of our AP measurements with the TM model only 1 degree of freedom is missing, which is the amplitude of the AP. At this point we fit the amplitude of the model to our experimental recording by estimating an appropriate extracellular conductivity.

The fast switch from resting to excited state in the generation of an AP is followed by a slow process of repolarization that restores the resting state. In order to provide a more general model that also accounts for repolarization, we add a simple extension to the TM model and refer to this as RTM. We approximate the repolarization in the extracellular potential by extending the TM model with an (ad hoc) exponential repolarization function (see Materials and methods).

## Comparison of measurements with model simulations

The above collision sweep experiment provides a detailed survey of the extracellular electric signature of APs. We now compare these results to the TM model, the RTM, and the classical HH model (see Materials and methods) by simulating a nerve of length 10 cm with 2000 compartments (conductivities in the HH model are area specific, here, we use a diameter of 80 µm). All models are adjusted to the MGF with identical procedure. The parameters $r_i$ and $c_m$ are used to match the propagation velocity $v_p$ and the width $\lambda^{\star}$ of the collision to the experimentally determined values. The fitting procedure is as follows: (i) Propagating APs are simulated. For a given $r_i$ the velocity is adjusted using $c_m$. (ii) An AP collision is simulated and the width is compared to the experiment (see *Figure 1—figure supplement 2*). If the width deviation is below 1%, the model is accepted. Otherwise $r_i$ is updated and the procedure is repeated from step (i). The TM model obeys the analytical expressions and converges immediately. The RTM deviates from the TM expressions only by a few percent and converges very fast. The HH model takes more iterations to converge. In all cases, the procedure is unambiguous and stable. The complete parameters for all models are given in *Appendix 3—table 2*.

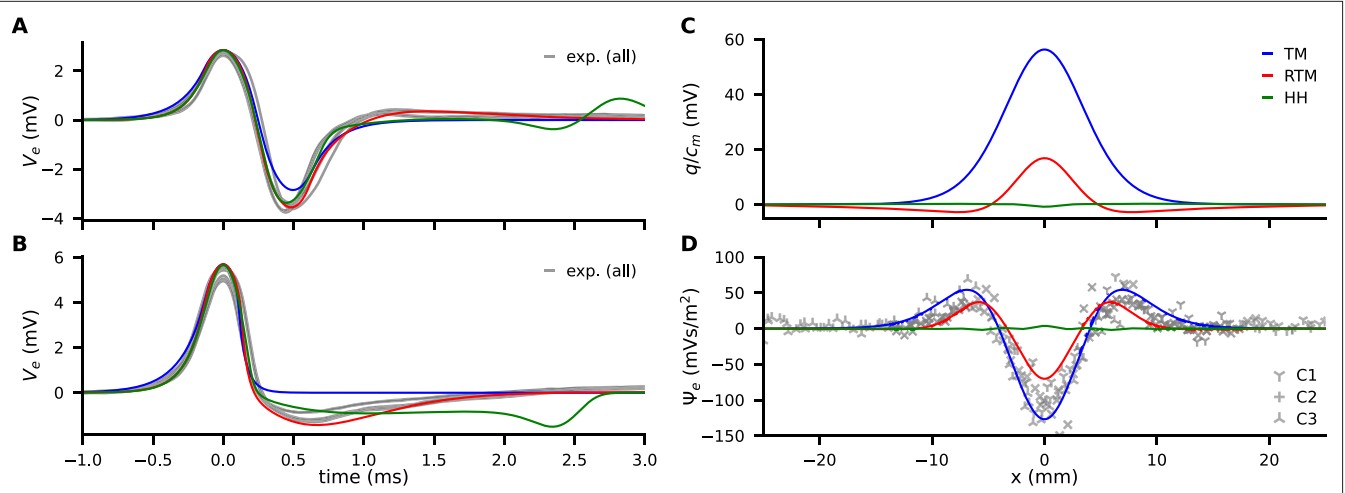

**Figure 3.** Comparison of experimental data (gray) with models (color, see legend). (**A**) Traces of $V_e(t)$ from propagating action potentials (APs). (**B**) Traces of $V_e(t)$ at the collision site. (**C**) Model predictions of the discharge ($q(x)$) generated by AP annihilation and released around the collision site. (**D**) The extracellular discharge of annihilating APs, measured and calculated. The gray crosses are the experimental values from four recordings at three separate sites along the median giant fibers (MGF), where each recording consists of up to 46 traces with varying delay times.

While the individual values of $r_i$ and $c_m$ depend on the choice of $g^\star$, the product $r_i c_m$ does not. From our measurements we obtain $r_i c_m = 26\,\text{s/m}^2$, which is in good agreement with literature values (**Tasaki and Matsumoto, 2002**; **Tasaki, 2012**). The resulting length parameter is $\lambda^\star = 1.8\,\text{mm}$ which is about half the width of the peak in **Figure 1D**. The deviations between the TM model and the experiment are: (i) Propagating APs cause a negative peak greater than the positive peak (**Figure 3A**). (ii) There is a small but sustained negative phase at the collision site (**Figure 3B**). The repolarization time in the RTM is found by comparing the model to the experimental traces of the colliding AP (**Figure 3B**). A value of $\tau_r = 0.5\,\text{ms}$ provides an acceptable fit. All models allow the reconstruction of the extracellular field of propagating APs within satisfying accuracy.

## Discharge of colliding APs

In an active axon, current always flows in and out in equal amounts (electroneutrality) (**Barbour, 2020**). As long as APs propagate the current must sum up to zero in space at any time and in time at any position. However, AP propagation is blocked when the axial current $I_{ax}$ is shut down at a boundary condition, e.g., by reaching the axon terminal or by AP collision. Such annihilating APs generate an effective net charge expelled into the extracellular space because the local sum in time can deviate from zero while the spatial sum remains zero. The associated current pulse may act like a stimulating electrode and as current source for ephaptic coupling. In order to access how much charge is expelled, we can calculate the charge as the integral of membrane current over time.

The membrane current $i_m$ is given by the cable equation (**Equation 1**) as the curvature of the membrane potential $\partial_x^2 V_m$, divided by $r_i$. Thus, the discharge $q(x)$ is the total charge expelled at a boundary (site of collision or axon terminal) and described by

$$q(x) = \int \mathrm{d}t\, i_m(x, t) = \frac{1}{r_i} \int \mathrm{d}t\, \partial_x^2 V_m(x, t) \tag{3}$$

The discharge $q(x)$ can be calculated with our models and yield very different predictions about its magnitude (**Figure 3C**). It has to be noted that $q(x)$ does not directly predict ephaptic coupling. However, $q(x)$ generates a curvature in the extracellular potential and in order to quantify possible impact of current, we integrate the curvature in the extracellular potential over time and define ($\Psi$) as

$$\Psi(x) = \int \mathrm{d}t\, \partial_x^2 V_e(x, t) \tag{4}$$

$V_e$ and thus $\Psi$ can be assessed with our collision measurements and can be used for model validation by comparing the models' predictions. In our simulations with a multi-compartment model, $\Psi$ is calculated when APs are colliding, based on parameterization from propagation velocity ($v_p$) and width of the collision (length $\lambda^\star$, see above). *Figure 3D* illustrates that the TM model is in excellent agreement with the experiment and the RTM only slightly underestimates $\Psi$. From this observation we conclude that the discharge is driven by the rapid onset of the AP and that the contribution by the repolarizing phase is small. On the other hand, the HH model largely disagrees with the experiments both in the magnitude of the discharge and in its temporal shape.

## Model of ephaptic coupling

First, we briefly investigate the effects of propagating APs on neighboring parallel cells which may give rise to synchronization (*Katz and Schmitt, 1940*; *Kriebel et al., 1969*; *Goldwyn and Rinzel, 2016*). Then, we focus on synaptic terminals, i.e., situations where the (finite) electric discharge of *annihilating* APs may influence surrounding target neurons.

We showed above that colliding APs emit a charge $q(x)$ and can act like a stimulating electrode. Within any cable model, the same boundary condition with $I_{ax} = 0$ also exist at axon terminals (*Plonsey, 1977*; *Spach and Kootsey, 1985*; *Kléber and Rudy, 2004*). It was our original idea to conduct the collision experiments and apply the multi-compartment simulations to then transfer the knowledge to annihilating APs at axon terminals. Likewise to our simulations for compartments around the site of collision, we can calculate the extracellular potential around the last compartments at the axon terminal. Knowledge of the generated inhomogeneity of $V_e$ allows to calculate the response of a target neuron that may be close to a site of AP annihilation. The details of this calculation are provided in the Materials and methods section.

Briefly, the above defined discharge from annihilation at terminals or collisions generates a distortion of the extracellular potential $V_e$ which in turn induces current in nearby target neurons.

In *Figure 4* we show simulations of ephaptic coupling in three different geometries: parallel neurons, end-shaft synapses, and end-end synapses. We use common literature values for the inner resistivity $\rho_i = 1\,\Omega\text{m}$ and membrane capacity $c_m = 10\,\text{mF/m}^2$ (*Cole, 1975*; *Carpenter, 1975*; *Bekkers, 2013*). Estimates for the extracellular resistivity are rare and highly variable, in the range of 3–600 $\Omega\text{m}$ (*Weiss et al., 2008*; *Lindén et al., 2011*). We derive $\rho_e = 100\,\Omega\text{m}$ by an estimate based on *Blot and Barbour, 2014*, as explained in Appendix 1.

The peak membrane conductivity in the TM model and RTM is $g^\star = 450\,\text{S/m}^2$, and the HH model is used with all classical parameters as specified in the supplementary material (e.g. $\bar{g}_{Na} = 1200\,\text{S/m}^2$). *Figure 4A* shows the potential $V_m(t)$ of the target neuron at the position marked by an arrow when excitation of a propagating AP occurs in the source neuron above. While the response in the target neuron calculated by the HH model is weak and broad, it is more asymmetric, peaked, shorter, and stronger in the TM model.

Next, we consider an axon terminal that ends near a neighboring, parallel target fiber. The chosen geometry as sketched in *Figure 4B* is a coarse representation of the synapse between Basket cell and Purkinje cell with the synapse embedded in a highly resistive structure, called the Pinceau (or Basket) at the axon initial segment (AIS). The TM model and RTM predict a strong and sharp initial hyperpolarization of the target neuron while the HH model predicts a comparably weak and smooth effect (*Figure 4B*, see also *Figure 4—figure supplement 1*).

In the RTM and HH model, this initial effect is followed by a more or less pronounced slow depolarization of the target neuron (*Figure 4C*). The RTM also reveals that a faster repolarization of the source AP increases the late depolarization of the target neuron.

The modulation of activity in Purkinje cells, e.g., changes in the rate of APs in response to an AP arriving at the Pinceau of the Basket cell was described in detail by *Blot and Barbour, 2014*. Since the relationship of membrane potential and the rate of APs in Purkinje cells is reported by the authors to be around $100\,\text{Hz/mV}$, we can compare our model predictions to the measured rate modulation of APs. We mapped the result of the models' predictions for different morphologies to the experimental data (*Blot and Barbour, 2014*) of AP rate modulation in Purkinje cells (*Figure 5*). Similarly to *Blot and Barbour, 2014*, we used a Gaussian filter with $\sigma = 1\,\text{ms}/2\sqrt{2}$. In our two models (TM and RTM), the modulation of not terminating but propagating APs along the source axon on the AP rate of the target cell is minute (*Figure 5A*). Note that this geometry does not correspond to the Purkinje

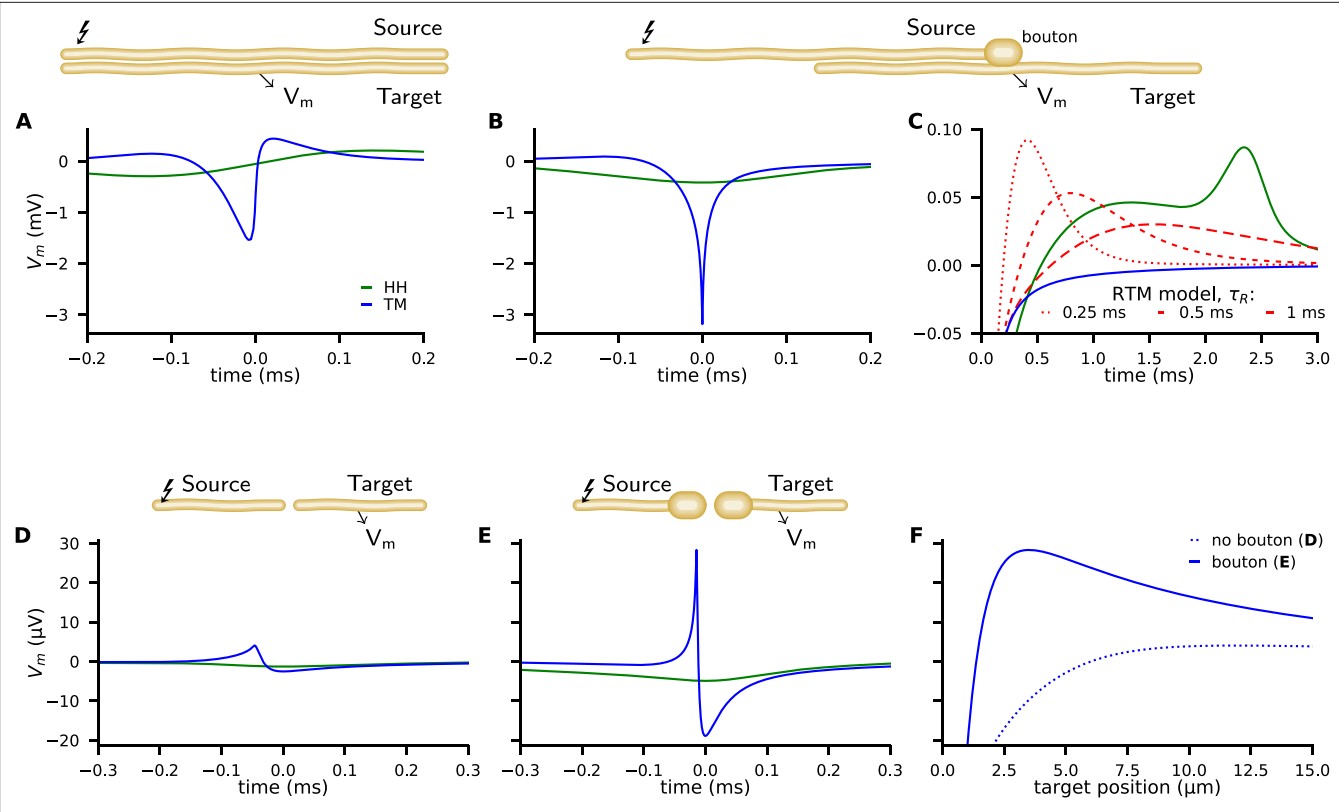

**Figure 4.** Examples of ephaptic coupling. Ephaptic coupling was calculated with the Tasaki-Matsumoto (TM) model (blue) and the Hodgkin-Huxley (HH) model (green): (**A**) in a parallel target neuron when an action potential (AP) is propagating in the source; (**B, C**) when the AP is annihilating at a bouton of a neuron terminal (upper neuron in end-to-shaft geometry, similar to the Basket cell-Purkinje cell synapse). The source and target neurons are 1 μm in diameter and are separated by 1 μm (2000 compartments each, length 1 mm, bouton size 2 μm). The neurons are placed next to each other, that is to say the numeric point compartments are separated by 1 μm. Traces denote the target membrane potential next to the point of axon termination. The initial hyperpolarization effect may be followed by a subsequent depolarization, depicted by the RTM (red) for different relaxation times. (**D–F**) Ephaptic coupling in an end-to-end synapse, illustrating the enhanced ephaptic coupling, due to enlarged neuron terminals (boutons). Here, the source and target neurons are 100 nm in diameter (2000 compartments each, length 300 μm, bouton size 400 nm). The target neurons are 1/4 in length and N (500 compartments, length 75 μm). The TM model generates a distant depolarization. Traces in D and E show the membrane potential of the target at the point where the TM model provokes its maximal depolarization (corresponding to the peaks in F). Traces in F show the spatial profile of the membrane potential along the target, at the time of maximal depolarization (corresponding to the peaks in **D, E**).

The online version of this article includes the following figure supplement(s) for figure 4:

**Figure supplement 1.** Simulation of target potential in an end-shaft synapse.

**Figure supplement 2.** Simulation of target potential in an end-end synapse.

cell-Basket cell connectivity. For annihilating APs at the axon terminal, with excitable segments up to the very end, our models reveal a moderate modulation, and only about half of what was reported for the Purkinje cell by *Blot and Barbour, 2014*. This illustrates the importance of AP annihilation for ephaptic coupling (*Figure 5B*). The Basket cell axon branches, forming the basket, and the branchlets or Pinceau embrace the initial part of the Purkinje cell axon. At the Pinceau a different set of ion channels is expressed and it is considered as being non-excitable (*Laube et al., 1996*; *Southan and Robertson, 2000*). We implemented the morphology of the Pinceau (increased volume and surface area) by implementing a bouton (diameter: axon 1 μm, bouton: 2 μm) and non-excitable terminal segments of 15 μm total length (*Bobik et al., 2004*). AP propagation is terminated before reaching the Pinceau terminal and the resulting modulation of AP rate in Purkinje cells highly depends on the morphology of the Basket cell and on $\tau$ of the RTM. Non-excitable segments at the last 15 μm result in very consistent predictions from both of our models (TM and RTM) for the reported inhibition and even the rebound activity (between 0.5 and 1.5 ms) is very well predicted by the RTM, using a $\tau$ of 0.5 ms (*Figure 5C*).

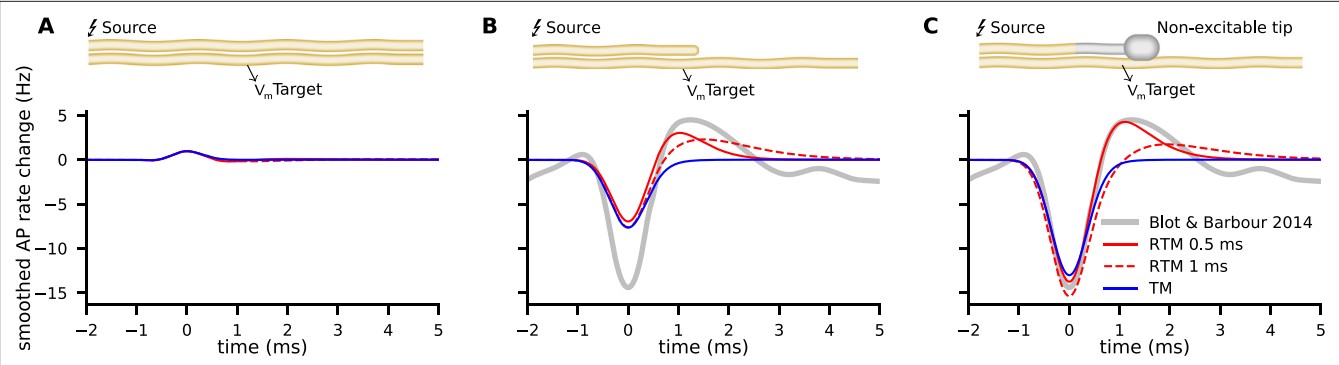

**Figure 5.** Comparison of experimental data from *Blot and Barbour, 2014*, on modulation of Purkinje cell activity (gray lines) with our relaxing Tasaki model (RTM) predictions (red lines with two different $\tau$) and the Tasaki-Matsumoto (TM) model prediction (blue) for three different geometries and physiological properties of the source neuron (e.g. the Basket cell). (**A**) Excluding the annihilation of an action potential (AP) at the source neuron, i.e., a propagating AP, cause only minute modulation of the predicted AP rate in the target neuron. Note that this example does not represent the Basket cell terminal with annihilating APs. (**B**) Annihilation of an AP at the terminal of the source neuron, with all segments being excitable in our calculation, cause moderate modulation. (**C**) Source neuron with similar properties to the Basket cell, i.e., a bouton and last segments non-excitable (corresponding to 15 μm with no switch from resting state to excited state), cause inhibition and rebound that is very similar as described by *Blot and Barbour, 2014*.

Beside the end-to-shaft geometry as in Basket cell-Purkinje cell connections, other morphologies and geometries are common in neural networks. We consider target neurons pointing toward the site of annihilation (*Figure 4D–F*, see also *Figure 4—figure supplement 2*). The geometry is reminiscent of an excitatory synapse, where an axon terminal is facing toward a dendritic spine. We choose the same parameters that we used for the end-shaft synapse, but the target neuron is placed in projection of the source nerve with a gap in between them of $10\,\text{nm}$. We include enlargements at the ends since boutons are common at the source and target neuron. Such enlargements significantly amplify ephaptic coupling.

We observe that the strong hyperpolarization of nearby postsynaptic membranes is accompanied by a distant depolarization. In *Figure 4D and E* we present the time course of the induced membrane potential in the target at the point of maximal depolarization. In *Figure 4F* the spatial profile of the induced voltage is given at different times.

The total amplitude of ephaptic coupling strongly depends on the choice of parameters. If we use $g^{\star} = 1200\,\text{S}\text{m}^2$ (as commonly used in the HH model) instead of the $450\,\text{S/m}^2$ used in *Figure 4*, the effect more than doubles. Note that much higher values up to $30\,\text{kS/m}^2$ have been reported for specific locations (*Holt and Koch, 1999*), which suggests even stronger ephaptic interactions in certain situations. The distant depolarization shown for the end-end synapse (*Figure 4D and E*) is only a few tens of μV, but as the amplitude of synchronized input from many dendrites adds up and cerebellar neurons have thousands of dendrites we may expect depolarization signal strengths in the mV range.

## Discussion

Our experimental design of a custom-made recording chamber with a nerve, hanging over a series of electrodes, allows accurate measurements of the space and time-dependent extracellular field $V_e$. Since, according to *Equation 9*, variations of $V_e$ mirror variations of the membrane current $i_m$, we obtain information on $i_m$, on the associated velocity $v_p$ and length scale $\lambda^{\star}$ of APs. Both of them are measured with great accuracy (about 10%), which allows us critical benchmarking of current theoretical models of APs. We find that both the TM model and the HH model are well suited to describe the extracellular electric potential of propagating APs which have symmetric forward and backward flow of $i_m$.

However, when an AP encounters a boundary, either at a axon terminal or in a collision with a counter-propagating AP, the current flow and $V_e$ become asymmetric. In fact, AP propagation is terminated when the overall local axial current is annihilated by a collision event and at axon terminals. At this point, the lack of internal stimulating current causes APs to disappear, as already described by *Tasaki, 1949*. During AP annihilation, the extracellular potential becomes a distinct monophasic spike.

This phenomenon has been observed for artificially produced as well as naturally occurring collisions in various systems (*Spach et al., 1971*; *Steinhaus et al., 1985*; *Tasaki, 1949*; *Tasaki, 1955*).

We describe the annihilation process by the time integrated charge ($q$, *Equation 3*) which is expelled at boundary condition, e.g., the axon terminal. From our collision experiments, we extract the spatial shape of the generated extracellular potential. The impact of extracellular potentials that are applied with microelectrodes is often described with the term 'activating function' (*Rattay, 2008*). We extended this view by the integration of the naturally occurring 'activating function' of neurons over time ($\Psi$, *Equation 4*).

We find that the experimentally observed values of $\Psi$ and thus the underlying discharge $q$ can be predicted very well by the TM/RTM models, while it is drastically underestimated by the HH model. The excellent fit of the TM model suggests that the discharge of annihilating APs is predominantly driven by the rapid depolarization at the onset of AP generation. The failure of HH is in line with observations made in various studies, indicating that the rapidity of membrane dynamics is not well described by the HH model (*Cole and Moore, 1960*; *Baranauskas and Martina, 2006*; *Naundorf et al., 2006*). Our quantitative physical description of ephaptic coupling between neurons provides a theoretical framework for investigating biological systems in which the phenomenon of ephaptic coupling already has been described and to explore other neural connections. The general cable equation, linking field inhomogeneity to membrane current, applies to both source and target cell. This leads to a coupling between source and target, driven by the electric field and its inhomogeneities. The cable model directly explains why AP-induced membrane current in a neuron induces membrane current in targets (*Durand, 2014*; *Merrill et al., 2005*; *Tung, 2021*). Surprisingly, this straightforward concept has never been applied to ephaptic coupling at synapses. Virtually all numerical computations of ephaptic coupling rely on the HH model which, as shown above, is not suitable to predict the effects caused by AP annihilation.

Electrical fields of propagating APs create positive and negative currents in equal amounts and a propagating AP first hyperpolarizes, then depolarizes a parallel aligned target neuron. This may cause synchronization of APs and our proposed model can also be used to study the observed phenomena of synchronization due to ephaptic coupling, even in the case of zero discharge (see *Figure 4A* and local impact on the target, integrated on timescales $> 1\,\mathrm{ms}$ in *Figure 5A*).

In contrast, AP annihilation in the source neuron acts as a strong local current injection. The nonzero discharge causes a more pronounced ephaptic coupling effect. AP annihilation occurs at the axon terminal and we have investigated ephaptic coupling for two neural connections to a greater extent. Both the pre- and postsynaptic neurons have similar morphology and orientation and, in addition, accessory structures at the site of connection. In teleost fish, an interneuron with an axon cap can have inhibitory, ephaptic coupling on the Mauthner cell's AIS, where APs are initiated (*Korn and Faber, 1975*; *Korn and Faber, 2005*).

In the vertebrate cerebellum, the Basket cells show ephaptic coupling with the Purkinje cell, and, comparably to the Mauthner cell, the generation of APs at the AIS is inhibited by a fast gating mechanism. In the following, we use the Basket cell-Purkinje cell connection as reference and for comparison to our model(s). As in all spiking neurons, an AP in the Basket cell generates a positive extracellular potential, and it is well documented that the Purkinje cell is very sensitive to extracellular potentials, since already $200\,\mathrm{V}$ modulates its firing rate (*Blot and Barbour, 2014*). The cap structure (Pinceau or Basket; *Ramón y Cajal, 1909*) around the terminal of the Basket cell restricts local currents to the Purkinje cell, increasing its sensitivity at the AIS to extracellular potentials (*Blot and Barbour, 2014*). The TM model predicts a fast and strong inhibition when APs are annihilating at the Basket cell (*Figure 4B*). Our numerical simulation revealed that in addition to the inhibitory effect of an annihilating AP at onset, a delayed ($0.5\,\mathrm{ms}$ later) depolarization can occur at the target neuron. The magnitude of this depolarization depends on the time course of AP repolarization at the source (*Figures 4C and 5*). Presynaptic variation of AP width is a widespread mechanism for modulation of synaptic transmission by changing transmitter release, and ephaptic coupling possibly contributes to a direct modulation at the postsynaptic neuron (*Begum et al., 2016*; *Southan and Robertson, 1998*; *Kole et al., 2015*). *Blot and Barbour, 2014*, described a biphasic modulation of Purkinje cells with delayed and synchronized APs in Purkinje cells after ephaptic inhibition by the Basket cells. In their study, the drug GABAzine was used to investigate the controversially discussed role of GABA for this modulation (*Iwakura et al., 2012*). However, a biphasic modulation is consistent with the purely electrical

framework provided in our RTM, and it remains to be investigated whether presynaptic modulation of AP width is the underlying mechanism for modulating the biphasic ephaptic coupling of Purkinje cells.

Our finding and formal description of the strong ephaptic coupling generated by annihilating APs impose the need and provide the possibility to examine bioelectric effects in other areas. Endogenous electric fields can influence molecular processes within cells, leading to cell growth, maturation, migration, and regeneration (*McCaig et al., 2000*; *Levin et al., 2017*; *Funk, 2015*; *Lyckman and Bittner, 1992*). The orientation of molecules and resulting structures can be induced by homogeneous electric fields and dielectric molecules can accumulate in field inhomogenities by dielectrophoresis (*Cifra, 2012*; *Pokorný, 2001*). In principle, such effects can lead to persistent structural changes in neurons and, thus, may contribute to neural plasticity and memory. During synapse formation, presynaptic neurons interact with spines of the prospective postsynapse. Prior, or in parallel, to chemical communication between the neurons that will subsequently form a synapse, Hebb's rule might be implemented by a discharge from the source and coincident activity in the target. In this case, ephaptic coupling might be instructive for synapse formation. We included calculations on ephaptic coupling with the geometry of an end-to-end synapse and boutons on source/target (*Figure 4C–E*) and we find that there is an initial sharp depolarization, followed by a slight hyperpolarization. Ephaptic coupling is highly amplified when source and target neurons have boutons, and such a morphology is omnipresent in spines and also presynaptic terminals commonly have enlargements as well. Our calculation further makes clear predictions where at the target (spine) depolarization can be expected, and this is the case about 1 μm away from the very tip of the target. It is important to highlight the importance of source-target geometry, predicted by our TM/RTM models, with opposite effects of ephaptic coupling in the two configurations: end-to-shaft and end-to-end.

Besides the annihilation of APs at the axon terminal, bidirectional propagation and hence collisions of APs might be more common than previously assumed (*Mateus et al., 2021*; *Scott et al., 2007*; *Debanne et al., 2011*). In theory, collisions can be used to perform computations (*Siccardi et al., 2016*) and neural networks possibly also perform such kind of information processing. For example, neurons for sound source localization perform a timing analysis based on binaural input. A neural network described as Jeffress delay line can accomplish such computation (*Jeffress, 1948*). *Franken et al., 2021*, performed a delay sweep experiment and their findings on integration at MSO neurons (medial superior olive neurons in vertebrates) look intriguingly similar to our collision experiment. Time differences and coincidence, in this case, is mapped on a location, e.g., the point of collision where a discharge with a center-surround profile is generated (*Figure 3D*; *Treue, 2014*).

Irrespective of such speculative functionalities, ephaptic coupling is ubiquitous. Its effects span spectacular length and timescales, in some cases it can bridge up to hundreds of microns (*Kriebel, 1968*; *Chiang et al., 2019*; *Shivacharan et al., 2019*).

Our formal description of ephaptic coupling between neurons provides a framework to study the functional significance of electric fields as a general mechanism for information processing in neural networks.

## Materials and methods
### Experimental design
The objectives of the study were (i) to measure the electric field around propagating and colliding APs with unprecedented accuracy; (ii) to benchmark a powerful yet simple model of APs; (iii) apply this model to demonstrate its predictive power for ephaptic coupling in general.

### Sample preparation
The experiments were performed with the VNC of earthworms (*Lumbricus terrestris*). The specimen is placed in anesthesia (0.2% butanol in tap water) for 20 min. Then, it is pinned in a basin, ventral side facing upward, and covered with preparation saline (0.04% butanol in saline, 26 mM NaSO$_4$, 25 mM NaCl, 6 mM CaCl$_2$, 4 mM KCl, 1 mM MgCl$_2$, 55 mM sucrose, 2 mM Tris, adjusted to pH 7.4; *Drewes and Pax, 1974*). The dissection starts with a small lateral incision caudally of the clitellum and it is followed by two longitudinal cuts alongside the VNC down to the posterior end. Afterward the middle lappet is removed to lay open about 10 cm of the VNC. Gently pulling up the nerve cord reveals lateral

connections which are then cut to disconnect the VNC from the rest of the nervous system. Once completely disconnected, the VNC is placed in chilled saline and kept at 4°C for about 1 hr.

## Electrophysiological recording of colliding APs

We use a custom-made nerve chamber made from polyoxymethylene which is encapsulated in a temperature-controlled aluminium case (cover not shown, *Figure 1B*) and kept at 12°C. It contains a row of silver chloride electrodes (diameter $0.8\,\text{mm}$), located $8\,\text{mm}$ above the bottom of the chamber and separated by $5\,\text{mm}$ from each other. The recording chamber is prefilled with chilled saline. The isolated VNC is transferred to the chamber and the saline is drained, leaving the VNC resting on the electrodes. Finally, the chamber is sealed with a plastic sheet and the aluminium case is closed. Two stimulators (Grass SD-9) are connected to electrodes at the ends of the nerve. In between, three custom amplifiers connect the recording electrodes to a digital storage oscilloscope (LeCroy WaveRunner 6050). Triggering of stimulation and recording is done with an Arduino microcontroller. Data is acquired with $1\,\text{MHz}$, smoothed with a Savitzky-Golay filter (width $51\,\mu\text{s}$), and baseline-corrected with the asymmetric least squares method (*Eilers, 2005*). The VNC contains two giant fibers, the MGF and LGF. These are unambiguously distinguished by their individual stimulation threshold and propagation velocity. For the analysis we used three collision sweep experiments. Each experiment consists of numerous recordings from three channels and varying delay, the complete data of experiment 3 is shown in *Figure 1—figure supplement 1*.

## Models

### General cable model for ephaptic coupling

The connection between inner electric potential $V$ and transmembrane current is given by the general cable equation (*McNeal, 1976*; *Rubinstein and Spelman, 1988*; *Rattay, 1999*; *Anastassiou et al., 2010*).

$$i_m = \frac{\partial I_{ax}}{\partial x} = \frac{\partial}{\partial x}\left(\frac{1}{r_i}\frac{\partial V}{\partial x}\right) = \frac{1}{r_i}\frac{\partial^2 V}{\partial x^2} + \frac{\partial V}{\partial x}\cdot\frac{\partial}{\partial x}\frac{1}{r_i} \tag{5}$$

This equation describes both the generation of membrane current caused by an AP and the concomitant current induced by an external potential, as used in clinical applications (*Durand, 2014*; *Merrill et al., 2005*; *Tung, 2021*). In an infinite homogeneous neuron the membrane current is determined by the second derivative of the potential. The second term accounts for the spatial change of resistivity at any structural inhomogeneities, varicosities, or neuron endings (*Basser and Roth, 2000*; *Holt and Koch, 1999*). Assuming a constant capacity and neglecting external fields and inhomogeneities in the neuron, one obtains the cable equation, *Equation 1*.

We assume the neurons being embedded in a large, homogeneous and isotropic conductor. In this case, the extracellular potential at a point $r$ that is generated by the source neuron is given by

$$V_e(r) = \rho \int \mathrm{d}x\, \frac{i_m(x)}{|r - x|} \tag{6}$$

where $\rho$ is the resistivity of the extracellular medium. The external potential $V_e(x,t)$ adds to the membrane potential $V_m(x,t)$ of the target neuron via $V = V_m + V_e$. Note that the target is considered to be in the resting state, where the transmembrane conductivity is negligible. The response is a redistribution of internal charges. In a static potential, the target neuron approaches a steady state by mirroring the external potential inhomogeneity in its transmembrane potential $V_m(x)$.

### Relaxing Tasaki model

The repolarization of the extracellular potential missing in the TM model is added ad hoc by a repolarization function as follows. We introduce a state parameter $n$, which is 1 for the resting state. When the membrane voltage crosses the threshold value, $n$ is set to 0 and its subsequent dynamic is given by

$$\dot{n}(t) = (1 - n(t))/\tau_r, \tag{7}$$

where $\tau_r$ is the repolarization time. The membrane parameters are controlled by

$$V_{eq} = -100\,\text{mV}/n^4 \quad \text{and} \quad g = g_m^\star(1 - n^4). \tag{8}$$

For $\tau_r \gg \lambda^\star/v_p$, the RTM reproduces the TM model, but eventually the resting state is restored a long time after an AP. With decreasing $\tau_r$ the repolarization affects the extracellular current and also the process of propagation. The exponent of $n^4$ effectively causes a delay of the repolarization. The values $g$ and $V_{eq}$ remain closer to the excited values for $t' < \tau_r/2$, where $t'$ is the time since excitation. Under this condition the influence of the repolarization upon the initial rising phase is negligible. Consequently, the TM expressions for $v_p$ and $\lambda^\star$ given above are in good agreement with the extended model (RTM).

## HH model

The celebrated HH model also relies on the cable equation but adds a couple of additional equations to mimic the spatiotemporal shape of an AP (*Hodgkin and Huxley, 1952*). In particular, it incorporates the Nernst equilibrium and specific voltage and time-dependent ionic conductances, resulting in a large number (typically of order 20!) of free parameters. This provides an enormous flexibility to account for almost any shape of AP. Nevertheless, our study reveals that the TM model and RTM which do not incorporate ion-specific conductances are capable to fit experimental APs very well with only 3 degrees of freedom. We compare our data with the HH model for reference because of its widespread use and popularity. The complete equations and parameters are provided in Appendix 2.

## Simulations

### Fitting to experiment

For simulations of the APs we use the three models introduced above (TM, RTM, HH), all with the same general parameters and morphology. The Python module BRIAN (*Stimberg et al., 2019*) is used to simulate the nerve with a multi-compartment model using a fourth-order Runge-Kutta method with time steps of $0.1\,\mu$s. After simulating the AP, the extracellular potential at the electrodes is calculated.

In the nerve chamber, the nerve cord is hanging free between electrodes. The measuring electrode is surrounded by two ground electrodes which are $5\,$mm apart. All sections of the VNC that are in between the surrounding ground electrodes contribute to the measured potential according to

$$V_{el} = R_{el} \int_{-d}^{d} \mathrm{d}x\, i_m(x') \frac{d - x'}{2d},$$
(9)

where $x'$ is the distance to the point of contact with the recording electrode, $d$ is the length of the freely hanging nerve between two neighboring electrodes, and $R_{el}$ is the resistance of the nerve cord between electrodes. We estimate the touching section of the nerve to be $0.8\,$mm (here $x' = 0$) and, accounting for the sagging of the nerve between electrodes, we integrate over $d = 5.2\,$mm in our calculations. In our approach, the magnitude of the TM model is controlled by $g_m^\star \cdot R_{el}$. For easier comparison, we first adjust the TM model to the magnitude of the HH model, which is achieved with $g_m^\star = 7.5\,$mS/m. All parameters are presented in *Appendix 3—table 2*.

Velocities are calculated from the time of AP arrival with an effective distance between the recording sites of $12\,$mm. The width of the collision is measured by the full width at half maximum of the negative deflection of the extracellular potential, as shown in *Figure 1—figure supplement 2*. The detailed settings as well as the calculated velocities and width are presented in *Appendix 3—table 1*.

### Predicted ephaptic coupling at synapses

We extended the standard BRIAN library to calculate the effect of an external field upon the target neuron by implementing *Equation 5*. The complete data for the end-shaft geometry with and without bouton for all models is shown in *Figure 4—figure supplement 1*. Complete data for the end-end synapse is shown in *Figure 4—figure supplement 2*. We provide the full source code at https://osf.io/duyn3/.

## Data and materials availability

Source code and data have been deposited at the Center for Open Science, https://osf.io/duyn3/ and at GitHub, https://github.com/moritz-s/Pyoelectricity, (copy archived at *Schloetter, 2025*) under GPL-2.0 licence. The repository contains the following three data files. Each file contains a delay sweep experiment, that is numerous recordings (with varying delay between opposing stimulations) from

three channels (three electrodes along the nerve). All the data are used in *Figures 2* and *Figure 3*. Please refer to *ExperimentAnalysis.ipynb* for information about the internal structure of the files and an example of how to open and use the data.

**ret0018.npz (exp.$\tilde{1}$)** 21 recordings, delay varied from 0 to 4 ms.

**ret0019.npz (exp.$\tilde{2}$)** 21 recordings, delay varied from 0 to 4.5 ms.

**ret0020.npz (exp.$\tilde{3}$)** 46 recordings, delay varied from 0 to 4.5 ms (raw data shown in *Figure 1—figure supplement 1*).

The repository contains all source code files used in this publication:

**pyoelectricity.py** A collection of functions to calculate propagating and colliding APs, the generated extracellular field, and its influence on surrounding cells.

**ExperimentAnalysis.ipynb** Analysis of experimental data and model fitting. Generates *Figures 1*, *Figure 1—figure supplement 2*, and *Figure 3*.

**end-end.py** Calculates the examples of ephaptic coupling at end-shaft synapses.

**end-end-plots.ipynb** Generates the figures for parallel propagation (*Figure 4A*) and end-shaft synapses (*Figure 4B and C*).

**end-end-plots.ipynb** Calculates the examples of ephaptic coupling at end-end synapses. Generates the figures for end-end synapses (*Figure 4D, E, and F*).

**Pinceau.ipynb** Calculation and plot of the Pinceau synapse (*Figure 5*).

The repository contains the following additional examples and test cases that are not explicitly used in the manuscript:

**test-ExternalField.ipynb** A general example of how to calculate the impact of extracellular fields via *pyoelectricity.py*.

**standalone-ExternalField.ipynb** A general standalone explanation of how to implement extracellular fields with the *generalized activating function* in BRIAN.

**Example1.ipynb** A simple example demonstrating the use of *pyoelectricity.py* to calculate ephaptic interactions.

Please refer to the *Readme.md* file for further instructions on usage and structure of the provided source code files.

## Acknowledgements

We thank Mahlon Kriebel, Georg Raiser, Sabine Kreissl, Shamit Shrivastava, and Gerardo Alvarez for fruitful discussions. This work was supported by the Deutsche Forschungsgemeinschaft (DFG) in the frame of the R Koselleck project Ma 817/9 as well as from the Zukunftskolleg Konstanz.

## Additional information

### Funding

| Funder | Grant reference number | Author |
|---|---|---|
| Deutsche Forschungsgemeinschaft | R. Koselleck project Ma 817/9 | Georg U Maret |
| Universität Konstanz | | Christoph J Kleineidam |

The funders had no role in study design, data collection and interpretation, or the decision to submit the work for publication.

### Author contributions

Moritz Schloetter, Conceptualization, Data curation, Software, Formal analysis, Validation, Investigation, Visualization, Methodology, Writing – original draft, Project administration, Writing – review and editing; Georg U Maret, Conceptualization, Supervision, Funding acquisition, Methodology, Writing – original draft, Writing – review and editing; Christoph J Kleineidam, Conceptualization, Resources, Supervision, Funding acquisition, Methodology, Writing – original draft, Project administration, Writing – review and editing

## Author ORCIDs
Moritz Schloetter (ID) https://orcid.org/0000-0002-1264-7704
Georg U Maret (ID) https://orcid.org/0000-0003-4069-648X
Christoph J Kleineidam (ID) https://orcid.org/0000-0003-0671-1455

## Ethics

This study was performed in strict accordance with the Regional German Authorities (Regierungspräsidium Freiburg) according to the German animal protection regulations specified in TierSchG and TierSchVersV. The protocol for Lumbricus terrestris required no additional approval. All surgery was performed under Trichlorbutanol anesthesia, and every effort was made to minimize suffering.

Reviewer #2 (Public review): https://doi.org/10.7554/eLife.88335.4.sa1
Author response https://doi.org/10.7554/eLife.88335.4.sa2

## Additional files

### Supplementary files
MDAR checklist

### Data availability
Complete source code and data is available at https://github.com/moritz-s/Pyoelectricity, (copy archived at *Schloetter, 2025*) and https://osf.io/duyn3 under GPL-2.0 licence. Please refer to the README.md file for detailed information.

The following dataset was generated:

| Author(s) | Year | Dataset title | Dataset URL | Database and Identifier |
| --- | --- | --- | --- | --- |
| Schloetter M, Kleineidam C | 2025 | Annihilation of action potentials induces electrical coupling between neurons | https://osf.io/duyn3/ | Open Science Framework, duyn3 |

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

## Appendix 1

### The specific conductivity between the Basket and the Purkinje cell

The Pinceau is ensheathing the axon of the Purkinje cell. *Blot and Barbour, 2014*, report a resistance of $R_{Pinceau} = 300\,\text{k}\Omega$ from inside the Pinceau to the surrounding ground.

We can estimate the specific conductance ($\rho$ in $\Omega$m) by assuming an effective geometry.

The Pinceau area is reported to be 100–600 $\mu\text{m}^2$(*Zhou et al., 2020*), e.g., a surface of $200\,\mu\text{m}^2$ corresponds to a sphere of radius $4\mu\text{m}$.

For a spherical shell with outer radius $a = 4\,\mu\text{m}$ and inner radius $b = 3.5\,\mu\text{m}$ we find

$$\rho = R_{\text{Pinceau}} \frac{4\pi}{b^{-1} - a^{-1}} = 106\,\Omega\text{m} \tag{10}$$

which is close to the $100\,\Omega$m used by us. Please note that the numerical values used here are coarse estimates with large uncertainties, still the order of magnitude is well compatible with literature values.

# Appendix 2

## HH model

The HH model (*Hodgkin and Huxley, 1952*) is implemented using standard parameters and expressions. We use the source code based on an example provided with the BRIAN package (https://brian2.readthedocs.io/en/stable/examples/compartmental.hodgkin_huxley_1952.html, *Stimberg et al., 2019*). The total resistive current is given by

$$i_r = g_l(E_l - v) + g_{Na}m^3h(E_{Na} - v) + g_K n^4(E_K - v)$$

These three terms (channels) drive the membrane toward specific equilibrium values

$$E_{Na} = 115\,\text{mV}, \quad E_K = -12\,\text{mV}, \quad E_l = 10.6\,\text{mV}$$

The conductivities $g_{Na}, g_K, g_l$ are controlled by

$$g_{Na} = 1200\,\text{S/m}^2, \quad g_K = 360\,\text{S/m}^2, \quad g_l = 3\,\text{S/m}^2$$

and

$$\frac{\mathrm{d}m}{\mathrm{d}t} = \alpha_m(1 - m) - \beta_m m, \quad \frac{\mathrm{d}n}{\mathrm{d}t} = \alpha_n(1 - n) - \beta_n n, \quad \frac{\mathrm{d}h}{\mathrm{d}t} = \alpha_h(1 - h) - \beta_h h$$

with

$$\alpha_m = 1\,\text{kHz/exprel}\left(\frac{-v + 25\,\text{mV}}{10\,\text{mv}}\right)$$

$$\beta_m = 4\,\text{kHz}\ \exp(-v/18\,\text{mV})$$

$$\alpha_h = 70\,\text{kHz}\ \exp(-v/20\,\text{mV})$$

$$\beta_h = 1\,\text{kHz/exp}\left(\frac{-v + 30\,\text{mV}}{10\,\text{mV}}\right)$$

$$\alpha_n = 100\,\text{Hz/exprel}\left(\frac{-v + 25\,\text{mV}}{10\,\text{mV}}\right)$$

$$\beta_n = 125\,\text{Hz}\ \exp(-v/80\,\text{mV})$$

To avoid singularities, this implementation uses the function exprel which is provided by BRIAN (*Stimberg et al., 2019*) and is defined as

$$\text{exprel}(x) = \begin{cases} 1, & \text{if } x = 0 \\ \dfrac{\exp(x) - 1}{x}, & \text{otherwise} \end{cases}$$

# Appendix 3

## Additional experimental results

**Appendix 3—table 1.** Experiment details settings as well as measured velocity and width of each experiment.

|  | Exp. 1 | Exp. 2 | Exp. 3 | (Unit) |
|---|---|---|---|---|
| # Recordings | 21 | 21 | 46 |  |
| Delay range | 4 | 4.5 | 4.5 | ms |
| Delay step | 0.2 | 0.23 | 0.1 | ms |
| Sampling rate | 1 | 1 | 1 | MHz |
| Duration | 10 | 10 | 10 | ms |
| $v_p$C1 $\rightarrow$ C2 | 16.2 | 16.2 | 16.3 | m/s |
| $v_p$C1 $\rightarrow$ C2 | 15.0 | 15.2 | 16.2 | m/s |
| $v_p$C1 $\leftarrow$ C2 | 15.1 | 15.1 | 14.9 | m/s |
| $v_p$C1 $\leftarrow$ C2 | 15.0 | 15.1 | 14.9 | m/s |
| $v_p$C2 $\rightarrow$ C3 | 13.1 | 13.1 | 13.3 | m/s |
| $v_p$C2 $\rightarrow$ C3 | 13.2 | 13.1 | 13.4 | m/s |
| $v_p$C2 $\leftarrow$ C3 | 13.0 | 14.3 | 14.5 | m/s |
| $v_p$C2 $\leftarrow$ C3 | 13.1 | 14.6 | 14.8 | m/s |
| Width | 10.2 | 9.9 | 8.7 | mm |

**Appendix 3—table 2.** Model parameters.

For all models, we used the measured propagation velocity $v_p$ (14.9 m/s) and the width of the collision, which is described by $\lambda^\star$ (1.8 mm) in the Tasaki-Matsumoto (TM) and relaxing Tasaki model (RTM), in order to adjust the parameters $r_i$ and $c_m$. For the Hodgkin-Huxley (HH) model, we used literature values describing the different channels, their conductivities, and time constants. For the TM model, we used the value of $g^\star$ as given by *Tasaki, 2002*. In order to compare the TM and the RTM with the HH model, we adjusted $g^\star$ such that the extracellular potential of the action potential (AP) is comparable. Although the predicted amplitudes of the extracellular potentials are very different, the products $r_i c_m$ are very similar and in good agreement with literature values (see *Tasaki, 2002*).

| Model | $r_i$(M$\Omega$/m) | $c_m$($\mu$F/m) | $r_i c_m$(s/m$^2$) | $g^\star$(mS/m) | Amp. |
|---|---|---|---|---|---|
| TM | 2.6 | 9.94 | 26 | 113.1 | 1.00 |
| HH | 61.5 | 0.46 | 28 | – | 0.07 |
| TM$_{HH}$ | 39.8 | 0.65 | 26 | 7.5 | 0.07 |
| RTM$_{HH}$ | 75.3 | 0.47 | 36 | 7.5 | 0.04 |

