## [Editor Report · eLife Assessment]

This **important** study enhances our understanding of ephaptic interactions by utilizing earthworm recordings to refine a general model and use it to predict ephaptic influences across various synaptic configurations. The integration of experimental evidence, a robust mathematical framework and computer simulations **convincingly** demonstrate the effects of action potential propagation and collision properties on nearby membranes. The study will interest both computational neuroscientists and physiologists.

---

## [Referee Report · Reviewer #2 (Public review)]

In this study, the authors measured extracellular electrical features of colliding APs travelling in different directions down an isolated earthworm axon. They then used these features to build a model of the potential ephaptic effects of AP annihilation, i.e. the electrical signals produced by colliding/annihilating APs that may influence neighbouring tissue. The model was then applied to some different hypothetical scenarios involving synaptic connections. In a revised version of the manuscript, it was also applied, with success, to published experimental data on the cerebellar basket cell-to-Purkinje cell pinceau connection. The conclusion is that an annihilating AP at a presynaptic terminal can emphatically influence the voltage of a postsynaptic cell (the 'electrical coupling between neurons' of the title), and that the nature of this influence depends on the physical configuration of the synapse.

As an experimental neuroscientist who has never used computational approaches, I am unable to comment on the rigour of the analytical approaches that form the bulk of this paper. The experimental approaches appear very well carried out, and the data showing equal conduction velocity of anti- and orthodromically propagating APs in every preparation are convincing.

The conclusions drawn from the synaptic modelling are considerably strengthened by the data in Figure 5. Here, the authors' model - including AP annihilation at a synaptic terminal - is used to predict the amplitude and direction of experimentally observed effects at the cerebellar basket cell-to-Purkinje cell synapse (Blot & Barbour 2014). One particular form of the model (RTM with tau=0.5ms and realistic non-excitability of the terminal) matches the experimental data extremely well. The authors also include a convincing demonstration (Panel A) that a propagating but not annihilating AP has almost no effect on a neighbouring neuron's activity. Given that the authors' model of ephaptic effects can quantitatively explain key features of experimental data pertaining to synaptic function, the implications for the relevance of ephaptic coupling at different synaptic contacts may be widespread and important.

---

## [Author Response]

The following is the authors’ response to the previous reviews.

**Reviewer 1:**
The authors explain that an action potential that reach an axon terminal emits a small electrical field as it "annihilates". This happens even though there is no gap junction, at chemical synapses. The generated electrical field is simulated to show that it can affect a nearby, disconnected target membrane by tens of microvolts for tenths of a microsecond. Longer effects are simulated for target locations a few microns away.To simulate action potentials (APs), the paper does not use the standard HodgkinHuxley formalism because it fails to explain AP collision. Instead it uses the Tasaki and Matsumoto (TM) model which is simplified to only models APs with three parameters and as a membrane transition between two states of resting versus excited. The authors expand the strictly binary, discrete TM method to a Relaxing Tasaki Model (RTM) that models the relaxation of the membrane potential after an AP. They find that the membrane leak can be neglected in determining AP propagation and that the capacitive currents dominate the process.The strength of the work is that authors identified an important interaction between neurons that is neglected by the standard models. A weakness of the proposed approach is the assumptions that it makes. For instance, the external medium is modeled as a homogeneous conductive medium, which may be further explored to properly account for biological processes. To the authors’ credit, the external medium can be largely varying and could be left out from the general model, only to be modeled specific instances.The authors provide convincing evidence by performing experiments to record action potential propagation and collision properties and then developing a theoretical framework to simulate effect of their annihilation on nearby membranes. They provide both experimental evidence and rigorous mathematical and computer simulation findings to support their claims. The work has a potential of explaining significant electrical interaction between nerve centers that are connected via a large number of parallel fibers.Comments on revisions:The authors responded to all of my previous concerns and significantly improved the manuscript.

We thank the reviewer for his comments and are pleased that we were able to adequately address all of his previous concerns. As a small comment to the remark of the reviewer “potential of explaining ... interaction ... via a large number of parallel fibers” we would like to add: The ephaptic coupling is prominent when APs annihilate at axon terminals, as we illustrate in Figure 4 and 5. Across parallel fibers, the impact of propagating APs is much lower but still may result in synchronization of APs.

**Reviewer 2:**
In this study, the authors measured extracellular electrical features of colliding APs travelling in different directions down an isolated earthworm axon. They then used these features to build a model of the potential ephaptic effects of AP annihilation, i.e. the electrical signals produced by colliding/annihilating APs that may influence neighbouring tissue. The model was then applied to some different hypothetical scenarios involving synaptic connections. In a revised version of the manuscript, it was also applied, with success, to published experimental data on the cerebellar basket cell-to-Purkinje cell pinceau connection. The conclusion is that an annihilating AP at a presynaptic terminal can emphatically influence the voltage of a postsynaptic cell (this is, presumably, the ’electrical coupling between neurons’ of the title), and that the nature of this influence depends on the physical configuration of the synapse.As an experimental neuroscientist who has never used computational approaches, I am unable to comment on the rigour of the analytical approaches that form the bulk of this paper. The experimental approaches appear very well carried out, and the data showing equal conduction velocity of anti- and orthodromically propagating APs in every preparation is now convincing.The conclusions drawn from the synaptic modelling have been considerably strengthened by the new Figure 5. Here, the authors’ model - including AP annihilation at a synaptic terminal - is used to predict the amplitude and direction of experimentally observed effects at the cerebellar basket cell-to-Purkinje cell synapse (Blot & Barbour 2014). One particular form of the model (RTM with tau=0.5ms and realistic non-excitability of the terminal) matches the experimental data extremely well. This is a much more convincing demonstration that the authors’ model of ephaptic effects can quantitatively explain key features of experimental data pertaining to synaptic function. As such, the implications for the relevance of ephaptic coupling at different synaptic contacts may be widespread and important.However, it appears that all of the models in the new Fig5 involve annihilating APs, yet only one fits the data closely. A key question, which should be addressed if at all possible, is what happens to the predictive power of the best-fitting model in Fig5 if the annihilation, and only the annihilation, is removed? In other words, can the authors show that it is specifically the ephaptic effects of AP annihilation, rather than other ephaptic effects of, say AP waveform/amplitude/propagation, that explain the synaptic effects measured in Blot & Barbour (2014)? This would appear to be a necessary demonstration to fully support the claims of the title.
**Reviewer 2 (Recommendations for the authors):**
Can you clarify whether all models shown in Fig5 involve an annihilating AP? Is it possible to plot the predicted effects of the most successful model (RTM 0.5ms in B) with *only* the annihilation selectively removed?

We are grateful for the reviewer’s comments and the specific suggestion for improvement (’...can the authors show that it is specifically the ephaptic effects of AP annihilation, rather than other ephaptic effects...’). For illustrating the importance of annihilation, we added the results of our calculation when no annihilation occurs, i.e. for propagating APs in the source neuron (Figure 5A) and we modified the geometry of the source neuron in Figure 5B such that only the annihilation takes place. Together with the source neuron with similar properties to the Basket cell (Figure 5C), we now show the effect of annihilation and the effect of Basket cell specific geometry and physiology. We added and edited in the main text the following 4 sentences:

ll 271: In our two models (TM and RTM), the modulation of not terminating but propagating APs along the source axon on the AP rate of the target cell is minute (Figure 5A). Note that this geometry does not correspond to the Purkinje cell-Basket cell connectivity. For annihilating APs at the axon terminal, with excitable segments up to the very end, our models reveal a moderate modulation, and only about half of what was reported for the Purkinje cell by Blot and Barbour (2014). This illustrates the importance of AP annihilation for ephaptic coupling (Figure 5B). We added and edited the figure legend:

Figure 5. ... (A) excluding the annihilation of an AP at the source neuron, i.e. a propagating AP, cause only minute modulation of the predicted AP rate in the target neuron. Note that this example does not represent the Basket cell terminal with annihilating APs. (B) annihilation of an AP at the terminal of the source neuron, with all segments being excitable in our calculation, cause moderate modulation. (C) source neuron with similar properties to the Basket cell, i.e. a bouton and last segments non-excitable (corresponding to 15 µm with no switch from resting state to excited state), cause inhibition and rebound that is very similar as described by Blot and Barbour (2014).

In the discussion, we extended one sentence to refer to Figure 5:

ll 346: This may cause synchronization of APs and our proposed model also can be used to study the observed phenomena of synchronization due to ephaptic coupling, even in the case of zero discharge (see Figure 4A, and local impact on the target, integrated on timescales >1 ms in Figure 5).